# Mechanically interlocked functionalization of monoclonal antibodies

Krzysztof P. Bzymek[1], James W. Puckett[1], Cindy Zer[1], Jun Xie[1], Yuelong Ma[1], Jeremy D. King[1], Leah H. Goodstein[1], Kendra N. Avery[1,4], David Colcher [2], Gagandeep Singh[3], David A. Horne[1] & John C. Williams [1]

Because monoclonal antibodies (mAbs) have exceptional specificity and favorable pharmacology, substantial efforts have been made to functionalize them, either with potent cytotoxins, biologics, radionuclides, or fluorescent groups for therapeutic benefit and/or use as theranostic agents. To exploit our recently discovered meditope–Fab interaction as an alternative means to efficiently functionalize mAbs, we used insights from the structure to enhance the affinity and lifetime of the interaction by four orders of magnitude. To further extend the lifetime of the complex, we created a mechanical bond by incorporating an azide on the meditope, threading the azide through the Fab, and using click chemistry to add a steric group. The mechanically interlocked, meditope–Fab complex retains antigen specificity and is capable of imaging tumors in mice. These studies indicate it is possible to "snap" functionality onto mAbs, opening the possibility of rapidly creating unique combinations of mAbs with an array of cytotoxins, biologics, and imaging agents.

[1] Department of Molecular Medicine, Beckman Research Institute of City of Hope, 1710 Flower St., Duarte, CA 91010, USA. [2] Department of Molecular Immunology, Beckman Research Institute of City of Hope, 1710 Flower St., Duarte, CA 91010, USA. [3] Department of Surgery, Beckman Research Institute of City of Hope, 1710 Flower St., Duarte, CA 91010, USA. [4] Present address: Xencor, 111W. Lemon Ave., Monrovia, CA 91016, USA. Correspondence and requests for materials should be addressed to J.C.W. (email: jcwilliams@coh.org)

We recently discovered a unique peptide binding site within a hole created by the light and heavy chains of the Fab domain of cetuximab[1], an anti-epidermal growth factor receptor monoclonal antibody (mAb) used clinically to treat head and neck and colorectal cancers (Fig. 1a). Because the position of the binding site lies within the middle of the Fab arm, we named the peptide, CQFDLSTRRLKC, that binds to this site a meditope. The residues that line the meditope binding site in the Fab are unique to cetuximab and not present in human mAbs[1]. Therefore, we hypothesized this site could be used as a unique 'receptor', not only for potentially attaching cargo[2,3], but also for emerging diagnostic techniques such as pre-targeted imaging[4]. Showing broad applicability of this technology, we successfully grafted the meditope site onto other mAbs, including trastuzumab, an mAb used to treat human epidermal growth factor receptor 2 (HER2)-positive breast cancer[1], and M5A, an anti-carcinoembryonic antigen (CEA) mAb[5]. We refer to mAbs onto which we have grafted the meditope site as meditope-enabled antibodies (memAbs). The affinity of the above memAbs for their cognate antigens is indistinguishable from that of the parental mAbs[1,6]. However, the half-life of the original meditope peptide–Fab complex is not optimal for a pre-formed memAb/drug-conjugated meditope combination to be successfully used in vivo. Although, mAbs can circulate in the body for days to weeks, the half-life of the original meditope–Fab interaction at 37 °C is only seconds. Herein, we introduce hydrogen bonds, increase the surface area, and eliminate strain to improve the half-life of the complex, allowing us to use click chemistry to sterically limit the dissociation of the meditope through the formation of a mechanical bond. We demonstrate that the mechanical bond permits the functionalization of a memAb, including the addition of fluorescent groups that permits the imaging of tumors in vivo.

## Results

**Improving the affinity of the interaction**. Based on previous studies on meditope/cetuximab complexes, we substituted positions 3 and 5 of the meditope with different side chains and variety of strategies for cyclizing the meditope[7,8]. We then used surface plasmon resonance (SPR) to characterize the affinity and kinetics of binding of these variants to a trastuzumab-based, anti-HER2 memAb (Supplementary Table 1). Of the modifications tested, substitution of Leu5 with L-diphenylalanine produced a significant increase in the affinity of the meditope to the anti-HER2 memAb, from 1.2 µM to 40 nM at 25 °C. Based on the kinetic fits, this increase was largely due to a faster on-rate (Fig. 1b and Supplementary Table 1). The structure of this modified meditope bound to the Fab of the anti-HER2 memAb indicated that the diphenyl group straddles Leu115 of the Fab heavy chain (Fig. 1c and Table 1), increasing the contact area by ~30 Å$^2$ [9]. Because diphenylalanine is a bulky, β-branched amino acid, part of the gain in affinity may reflect reduced entropy, which also contributes to a faster on-rate. In addition, we found that extending the C-terminus of the original meditope by four residues, Gly–Gly–Ser–Lys, to subsequently add a fluorophore through amine chemistry, also improved the affinity. These four residues, however, were disordered in the crystal structure of the extended meditope bound to the anti-HER2 memAb. Therefore, additional experiments are necessary to better understand the contribution of these residues to the increase in the overall affinity. A limited set of additional substitutions to the original meditope did not produce substantial gains in affinity (Supplementary Table 1).

Turning to the Fab, examination of the crystal structure of the meditope bound to the Fab of the anti-HER2 memAb revealed that the guanidinium group of Arg9 in the meditope is partially buried in a cavity lined with Ile83 of the Fab light chain. This suggested substitution with a negative charge at Ile83 would result in a favorable buried salt bridge. Mutation of Ile83 to glutamate substantially improved the affinity for the original meditope (i.e., the unmodified CQFDLSTRRLKC peptide) to 23 nM as determined by SPR (Supplementary Table 1). The crystal structure of the original meditope in complex with the Ile83Glu anti-HER2 meditope-enabled Fab confirmed the formation of a favorable electrostatic interaction with a Glu83-Arg9 bond distance $d_{OE2\cdots NH1} = 2.7$ Å (Fig. 1c). As with the other meditope/meditope-enabled Fab structures, the substitution did not substantially change the overall structure of the Fab (the root-mean-square-deviation between the Fab domains of the Ile83Glu variant and the parental trastuzumab calculated over Cα is 0.28 Å).

Combining these meditope (i.e., Leu5 to diphenylalanine substitution and C-terminal extension) and meditope-enabled Fab (i.e., Ile83Glu substitution) modifications produced a dramatic increase in the affinity. The binding constant could not be accurately determined at 25 °C because the off-rate exceeded the instrument's limits ($k_d < 10^{-5}$ s$^{-1}$). The upper limit on the affinity, based on the calculated on-rate ($k_a = 3.8 \times 10^{-5}$ M$^{-1}$s$^{-1}$ and $k_d = 10^{-5}$ s$^{-1}$ (instrumental limit)), was 26 pM. Compared to the starting affinity, these three modifications improved the affinity by more than four orders of magnitude. To more accurately quantify the increase in affinity, we conducted SPR experiments at 37 °C. The extended, 5-diphenylalanine meditope with the Ile83Glu Fab mutation produced a final $K_D$ of 0.40 nM (Fig. 1c, lower right panel), increasing the half-life of the interaction from 16 s to almost 40 min at 37 °C. Thus, at physiological temperatures, these three modifications collectively produced a 2500-fold increase in affinity (initial $K_D = 1000$ nM; final $K_D = 0.40$ nM, Fig. 1c).

**Mechanically interlocked meditope mAb complex**. Although considerably longer than the initial half-life of the meditope/meditope-enabled Fab complex, a half-life of 40 min is short compared to the half-life of mAbs in vivo, often greater than 14 days[10]. We observed in the structure of the meditope/meditope-enabled Fab complex that the Arg8 side chain of the meditope is partially accessible from the other side of the Fab hole (Fig. 2a). Therefore, we investigated if we could incorporate a non-natural amino acid with an orthogonal reactive group at this position, such that the side chain would extend through the hole of the meditope-enabled Fab and expose the reactive group to the other side of the Fab[11,12]. Then, exposing the modified meditope/Fab complex to a steric group that specifically reacts with the modified side chain would effectively lock the meditope onto the Fab (Fig. 2b). In other words, we sought to generate a mechanical bond to sterically limit the dissociation of a functionalized meditope from the Fab (Fig. 2c)[13].

To create the mechanical bond, we added a two- or a three-subunit ethylene linker with a terminal azide to the guanidinium group of arginine, which we then incorporated into the Arg8 position of the meditope. Using SPR, we found that this 5-diphenylalanine-8-Arg-(polyethyleneglycol)$_2$-azido-meditope bound to the Ile83Glu meditope-enabled Fab with similar affinity as the 5-diphenylalanine meditope ($K_D$ of 2.1 nM vs. 0.40 nM, respectively) but with a slower on-rate and off-rate, corresponding to a longer half-life ($\tau = 83$ min at 37 °C) (Supplementary Fig. 1a). Crystal structures of the 5-diphenylalanine-8-Arg-(PEG)$_2$- and 5-diphenylalanine-8-Arg-(PEG)$_3$-azido-meditope bound to the Ile83Glu meditope-enabled Fab showed clear electron density of the meditope as well as the PEG-azide. Important for creating the mechanical bond, the structures

indicated the azide was threaded through the Fab hole (Supplementary Fig. 1b, c and Table 1).

We locked the azido-meditope onto the meditope-enabled Fab by reacting the complex with dibenzocyclooctyne-AlexaFluor647 (DIBO-AF647). We then isolated the complex by size exclusion chromatography (SEC), monitoring the absorbance at 280 nm and at 647 nm. The signal from AF647 overlapped with the 280 nm signal from the meditope-enabled Fab, suggesting that the meditope was interlocked with the meditope-enabled Fab

(Supplementary Fig. 3). In control SEC experiments, the DIBO-AF647 dye alone did not co-migrate with the meditope-enabled Fab either in the absence of the azido-meditope or the presence of a non-azido-meditope. Although we have made multiple attempts at crystallizing the mechanically interlocked meditope/meditope-enabled Fab complex, the flexibility and stereochemical heterogeneity of the system have thus far precluded us from obtaining atomic resolution data of the interlocked complex. Therefore, to provide evidence that the click reaction did in fact produce a

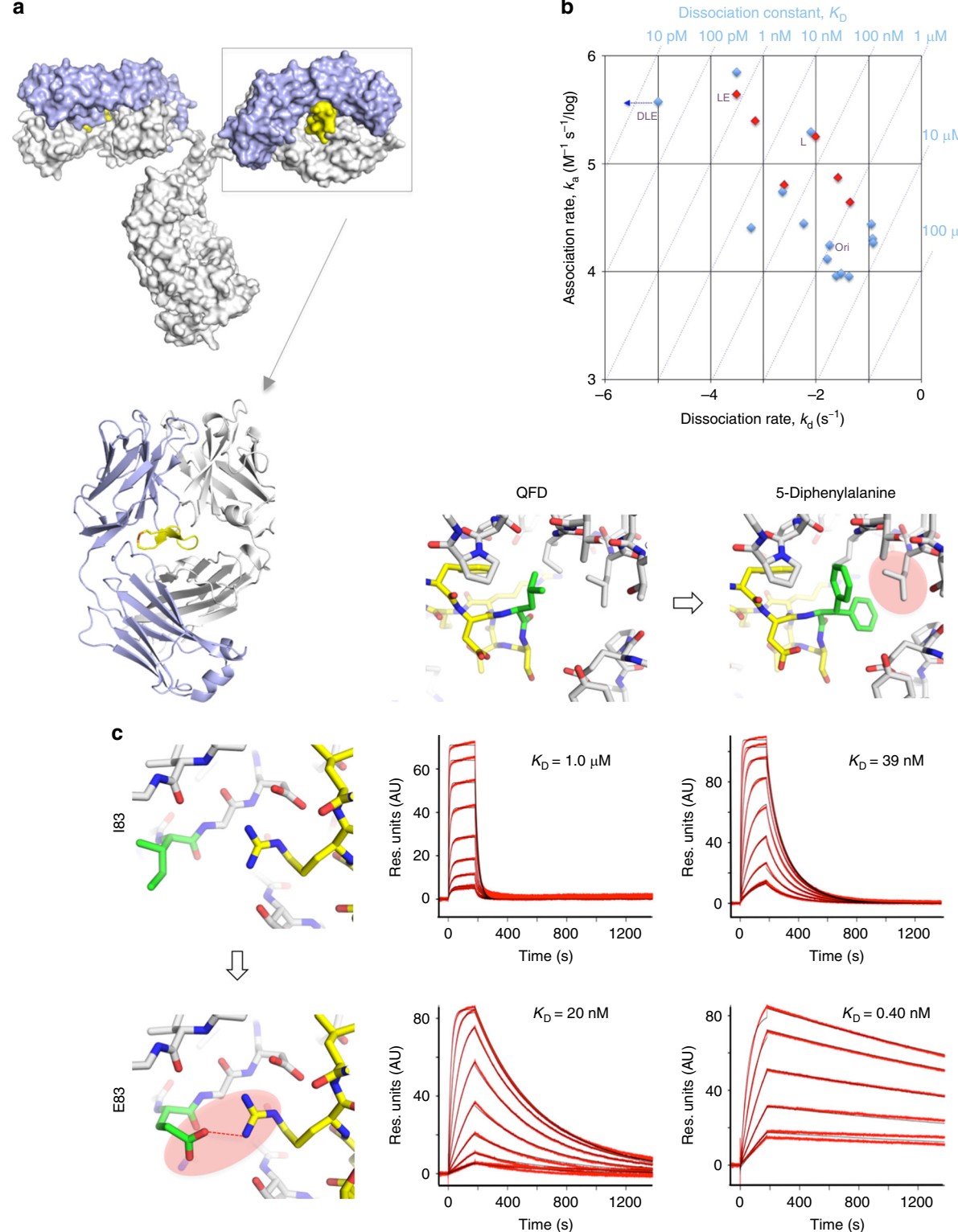

mechanically interlocked complex, we analyzed by mass spectrometry the mechanically interlocked, meditope/Ile83Glu anti-HER2 memAb complex under native conditions as well as the mechanically interlocked Ile83Glu meditope-enabled Fab/meditope complex under denaturing conditions (Fig. 2d and Supplementary Fig. 4). The calculated mass from native mass spectra of the interlocked azido-meditope–AF647/Ile83Glu anti-HER2 memAb was consistent with the formation of the mechanical bond on one or both Fab arms of the IgG.

To test the stability of the mechanically interlocked complex, we incubated the interlocked azido-meditope–AF647/Ile83Glu meditope-enabled Fab at 37 °C for 11 days and used SEC to follow the dissociation of the complex (Supplementary Fig. 5). At 37 °C, the complex was stable over 11 days, as determined by SEC as well as by flow cytometry with mechanically interlocked Ile83Glu meditope-enabled IgG complex (Supplementary Figs. 5 and 6). To show this approach is generalizable to other mAbs, we created meditope-enabled, anti-CD3 (Okt3), anti-HER2 (pertuzumab), and anti-CEA (M5A)[6] antibodies (i.e., IgG), and used the same click chemistry to mechanically interlock the azido-meditope with DIBO–AF647. Each mechanically interlocked meditope–memAb complex bound to its cellular target as determined by analytical flow cytometry, indicating the formation of the mechanically interlocked bond does not affect antigen binding in general (Supplementary Fig. 7).

Finally, we used click chemistry to demonstrate the possibility of adding functionalities other than fluorophores to the complex. In one instance, we interlocked a dibenzylcyclooctyne (DBCO)-His$_6$ moiety, which can facilitate purification of the mechanically interlocked complex from unreacted substrates (Supplementary Figs. 2 and 8d). We also conjugated the azido-meditope/meditope-enabled Fab complex to magnetic beads conjugated with DBCO (Supplementary Fig. 9). Finally, to create a "molecular dumbbell," we conjugated DBCO-(PEG)$_4$-NHS to an acetylated, seven amino acid peptide with lysines at the N- and C- termini and reacted this bis-DBCO linker with the azido-meditope/meditope-enabled Fab complex. Native mass spectrometry indicated the formation of a mechanically interlocked molecule (Fig. 2d and Supplementary Fig. 10). The combination of orthogonal conjugation chemistries (e.g., azide/alkyne[11,12], tetrazine/trans-cyclooctene (TCO)[14,15], etc.) and multivalent linkers opens the possibility of creating novel, bi-specific and multi-specific Fab/IgG complexes[16,17].

**Tumor imaging using the interlocked meditope IgG complex.** To demonstrate a potential application of our approach, we sought to use the interlocked meditope–mAb complex to image tumors in vivo in xenograft-bearing mice. First, we show using flow cytometry that the mechanically interlocked meditope–AF647/Ile83Glu anti-HER2 memAb complex bound to HER2-overexpressing BT474 cancer cells (Fig. 3a). As a control for the animal imaging studies, we also produced the mechanically interlocked meditope–AF647/Ile83Glu anti-CD3 memAb and demonstrated by analytical cytometry that this mechanically

interlocked complex binds CD3-bearing Jurkat cells (Fig. 3b). Of note, Okt3 binds to human CD3 but does not cross react with murine CD3. Next, the azido-meditope–AF647/Ile83Glu anti-HER2 memAb complex or azido-meditope–AF647/Ile83Glu anti-CD3 memAb complex was injected in the tail vein of NSG female mice (approximately 9 weeks old, Jackson Laboratory) bearing BT474 HER2-overexpressing breast tumor xenografts. The mechanically interlocked, AF647–meditope/Ile83Glu anti-HER2 memAb accumulated at the tumor within 24 h of injection and remained at 72 h albeit with less intensity (Fig. 3c, top row). There was no signal observed in three of the four mice using the azido-meditope–AF647/Ile83Glu anti-CD3 memAb control. A weak signal at the site of the tumor was observed in the fourth mouse (and in the tail), but was absent by 72 h. After the final images were collected at 72 h, the mice were euthanized and their major organs imaged. The fluorescent signal was found predominantly in tumors and residual signal was observed in the stomach (GI) using azido-meditope–AF647/Ile83Glu anti-HER2 memAb (Fig. 3d). No signal was observed in the tumors in the control group. However, there was some residual signal in the stomach in the controls, similar to the experimental group. These results were consistent with the initial imaging experiments (Supplementary Fig. 11).

## Discussion

To leverage their high specificity and favorable pharmacokinetics/pharmacodynamics, there has been and continues to be considerable interest to add functionality to monoclonal antibodies for therapeutic and diagnostic intent[18–21]. Invariably, each of these approaches involves chemical modification directly to the mAb. The approach developed here differs from these methods—the interlocked mechanical bond is sterically restricted and non-covalent by nature[22]. To achieve this, we needed to ensure the lifetime of the meditope-mAb complex was sufficiently long that the capping of the meditope using click chemistry would predominantly occur as part of the complex. Using a rational, structural-based approach, we significantly enhanced the affinity of the meditope/meditope-enabled Fab interaction. Next, we modified the guandinium group of arginine 8 within the meditope to thread an azide group through the Fab hole. Using copper-free, click chemistry, we were able to generate several mechanically interlocked meditope/mAb complexes. Functionalization of the mAbs/Fabs reported here did not affect their antigen-binding properties as evidenced by SPR (Supplementary Fig. 8) and/or analytical cytometry (Fig. 3a, b and Supplementary Fig. 7b and 7c). As an example of its potential use in the clinic, we demonstrate that the interlocked azido-meditope–AF647/Ile83-Glu anti-HER2 memAb complex can be used to effectively image tumors in mice. Overall, the data presented here, to the best of our knowledge, are the first example of mechanically interlocked complex involving an antibody.

In future studies, we will refine the parameters to optimize the efficiency of the interlocking reaction, including characterization of differing orthogonal chemistries (e.g., thiols and tetrazine/TCO)

---

**Fig. 1** Increasing the affinity of the meditope site. **a** Surface representation of an IgG with a bound meditope (yellow). Light blue indicates the light chain and white indicates the heavy chain. **b** Kinetics and thermodynamics of meditope and antibody modifications ($n = 1$). Blue points represent data collected at 25 °C. Red points represent data collected at 37 °C (see also Supplementary Table 1). DLE—5-diphenylalanine long meditope ((Ac)CQFDA (Ph)$_2$STRRLRCGGSK) binding to Ile83Glu anti-HER2 memAb; LE—long meditope ((Ac)CQFDLSTRRLRCGGSK) binding to Ile83Glu anti-HER2 memAb; L—long meditope ((Ac)CQFDLSTRRLRCGGSK) binding to anti-HER2 memAb antibody; ori—original meditope (CQFDLSTRRLKC) binding to anti-HER2 memAb. **c** SPR sensograms of meditope peptide variants binding to immobilized memAb variants at 37 °C ($n = 1$). Top, left sensogram—CQFDLSTRRLKC meditope (QFD, original unmodified meditope) binding to anti-HER2 memAb (original, meditope-enabled anti-HER2 antibody); top, right sensogram—(Ac) CQFDA(Ph)$_2$STRRLRCGGSK (5-diphenylalanine long meditope) binding to anti-HER2 memAb; bottom, left sensogram—QFD binding to Ile83Glu anti-HER2 memAb; bottom, right sensogram—5-diphenylalanine long meditope binding to Ile83Glu anti-HER2 memAb. Residues highlighted in green correspond to modifications in the meditope (position 5, top panel) or Fab (position 83 LC, left panels)

**Table 1 Data collection and refinement statistics**

| | Apo Ile83Glu Fab (5U3D) | Ile83Glu Fab+ 5-diphenylalanine long meditope[a] (5U5F) | Ile83Glu Fab+ 5-diphenylalanine-8-arginine-(PEG)$_2$ azido-meditope[a] (5U6A) | Ile83Glu+ 5-diphenylalanine-8-arginine-(PEG)$_3$ azido-meditope[a] (5U5M) |
|---|---|---|---|---|
| *Data collection* | | | | |
| Space group | P2$_1$2$_1$2$_1$ | P2$_1$2$_1$2$_1$ | P2$_1$2$_1$2$_1$ | P2$_1$2$_1$2$_1$ |
| *Cell dimensions* | | | | |
| a, b, c (Å) | 52.85; 104.65; 116.88 | 53.43; 105.38; 117.00 | 53.25; 105.17; 117.07 | 52.53, 105.47, 117.13 |
| α, β, γ (°) | 90.0, 90.0, 90.0 | 90.0, 90.0, 90.0 | 90.0, 90.0, 90.0 | 90.0, 90.0, 90.0 |
| Resolution (Å) | 33.43–1.77 (1.82–1.77)[b] | 33.64–1.81 (1.86–1.81) | 31.53–1.74 (1.78–1.74) | 31.62–1.88 (1.93–1.88) |
| $R_{mrgd-F}$ | 0.075 (0.760) | 0.085 (0.784) | 0.062 (0.485) | 0.100 (0.863) |
| $I/\sigma(I)$ | 18.6 (1.88) | 16.1 (1.66) | 15.6 (2.24) | 16.4 (2.12) |
| Completeness (%) | 99.3 (91.1) | 99.4 (93.7) | 95.6 (79.0) | 99.9 (99.8) |
| Redundancy | 4.5 (3.2) | 3.9 (2.8) | 2.5 (2.0) | 5.2 (5.1) |
| *Refinement* | | | | |
| Resolution (Å) | 1.77 | 1.81 | 1.74 | 1.88 |
| No. reflections | 63,311 | 60,648 | 65,699 | 54,735 |
| $R_{work}/R_{free}$ | 15.7/18.5 | 16.3/19.2 | 15.9/18.8 | 16.3/19.7 |
| *No. atoms* | | | | |
| Protein | 4350 | 4421 | 4467 | 4328 |
| Meditope/ligand | –/8 | 115/8 | 137/8 | 150/8 |
| Water | 803 | 798 | 857 | 732 |
| *B-factors* | | | | |
| Protein | 21.4 | 23.1 | 19.5 | 25.1 |
| Meditope/ligand | –/28.2 | 22.7/20.0 | 24.4/21.5 | 27.5/22.2 |
| Water | 35.7 | 37.2 | 35.2 | 37.6 |
| *R.m.s. deviation* | | | | |
| Bond lengths (Å) | 0.007 | 0.007 | 0.007 | 0.007 |
| Bond angles (°) | 1.048 | 1.075 | 1.098 | 1.082 |

[a]-5-diphenylalanine long meditope – (Ac)CQFDA(Ph)$_2$STRRLRCGGSK; 5-diphenylalanine-8-arginine-(PEG)$_2$-azido-meditope – (Ac)CQFDA(Ph)$_2$STRXLRCGGSK, where X is arginine-(PEG)$_2$-azide; 5-diphenylalanine-8-Arg-(PEG)$_3$-azido-meditope – (Ac)CQFDA(Ph)$_2$STRXLRCGGSK, where X is arginine-(PEG)$_3$-azide
[b]All data were collected from single crystals. Highest resolution shell is shown in parenthesis

and steric restraints. We will also create a toolbox of multiple meditope and DBCO analogs bearing cytotoxins (e.g., Auristatin), imaging agents (e.g., 1,4,7,10-tetraazacyclododecane-1,4,7,10-tetraacetic acid (DOTA) for radionuclide imaging studies), and other functionalities. One potential application would be the generation of bi-specific antibodies with tunable lengths between the complementarity determining regions of the Fabs to optimize antibody-dependent cellular cytotoxicity (ADCC), an example of which is the "molecular dumbbell". We anticipate this toolbox will be compatible with any memAb and allow for the rapid and consistent development of multiple functional biologic agents.

## Methods

**Reagents**. All reagents were of American Chemial Society (ACS) or higher purity and were purchased from Thermofisher or Sigma unless specified below. Cy5-DBCO, Cy5-N-hydroxysuccinimide (NHS), DBCO-(PEG)$_4$-NHS ester and sulfo DBCO-(PEG)$_4$-maleimide, and DBCO-magnetic beads were purchased from Click Chemistry Tools. Cell lines (BT474 and LS174T) were obtained from ATCC, validated by flow cytometry, and tested monthly to ensure they were free of mycoplasma contamination (Universal Mycoplasma Detection Kit from ATCC).

**Peptide synthesis**. Standard solid-phase N-αFmoc chemistry was used to synthesize meditope derivatives on CS136XT peptide synthesizer (CS Bio). Reagent K (trifluoroacetic acid (TFA)/water/phenol/thioanisole/1,2-ethanedithiol (EDT) = 82.5:5:5:5:2.5) was used to cleave the peptides from the resin. Crude peptides were collected by precipitation with cold ether. Oxidation using 20% DMSO in ammonium acetate buffer (pH 6) was performed for disulfide cyclization. All peptides were purified using a reverse-phase high-performance liquid chromatography (RP-HPLC) (Agilent 1200 system with Agilent prep-C18 column, 21.2 ×

150 mm, 5 µm) with a water (0.1% TFA)/acetonitrile (0.1% TFA) solvent system. Alexa647- and Cy5-labeled peptides were synthesized from Alexa647-NHS and Cy5-NHS and purified using RP-HPLC. All peptides were characterized by mass spectrometry.

**Synthesis of azido-meditope variants**. Automated peptide synthesis was used to generate 2.46 g HMP Resin-KSGGCRLR-Orn(Alloc)-TS-3,3-diphenylalanine-DFQC-NHAc upon drying, starting from 500 µmol HMP Resin-Lys(Boc)-NHFmoc, sourced from CS Bio. All monomers were sourced from CS Bio, except Fmoc-Orn(Alloc)-OH, which was purchased from Alfa-Aesar.

Azido-meditope variants were synthesized according to the literature[23]. Briefly, resin II-20 (454 µmol, Supplementary Methods) was subjected to 20% piperidine in dimethylformamide (DMF) to remove Fmoc (1 × 5 min, 1 × 20 min). The resin was rinsed 3 × 10 mL DMF, 3 × 10 mL dichloromethane (DCM), 3 × 10 mL DMF. The resin was acetylated using 10 mL 1:2:7 acetic anhydride:N,N-diisopropylethylamine (DIEA):DMF (v/v/v). The resin was again rinsed 3 × 10 mL DMF, 3 × 10 mL DCM, 3 × 10 mL methanol (MeOH). The resin was dried in vacuo, and 247.5 mg of resin (~50 µmol) was then subjected to dealloc conditions: Pd(PPh$_3$)$_4$ (14.5 mg, 12.5 µmol), 1.3-dimethylbarbituric acid (DMBA) (78 mg, 500 µmol), and 1:1 N-methylpyrrolidine (NMP):DCM (2 mL). The resin was copiously washed with DMF, DCM, DMF, 1 vol. 5% DIEA in DMF (by volume), 2 × 5% sodium diethyldithiocarbamate trihydrate in NMP, and then copious amounts of DMF, DCM, and MeOH (greater than 10 washed at approximately twice the volume of resin) to deswell the resin. The resin was dried in vacuo. Compound **a** (667 mg, 312 µmol) in 1.5 mL tetrahydrofuran (THF) and 20 µL triethylamine (TEA) was added to the dried resin. The reaction proceeded for 20 h at room temperature in a rocking, manual, solid-phase peptide synthesis vessel. The Kaiser test of a small portion of the resin was positive. Compound **a**[23] was reconcentrated and dissolved in DCM (1.5 mL). The coupling was restarted in the presence of DCM and 30 µL TEA. After 22 h, the Kaiser test was still positive, and semi-preparative HPLC of a small fraction of material showed ~50% conversion. The resin was rinsed 2 × DMF, 2 × DCM, 2 × DMF, 2 × DCM, deswollen in MeOH, and then dried in vacuo.

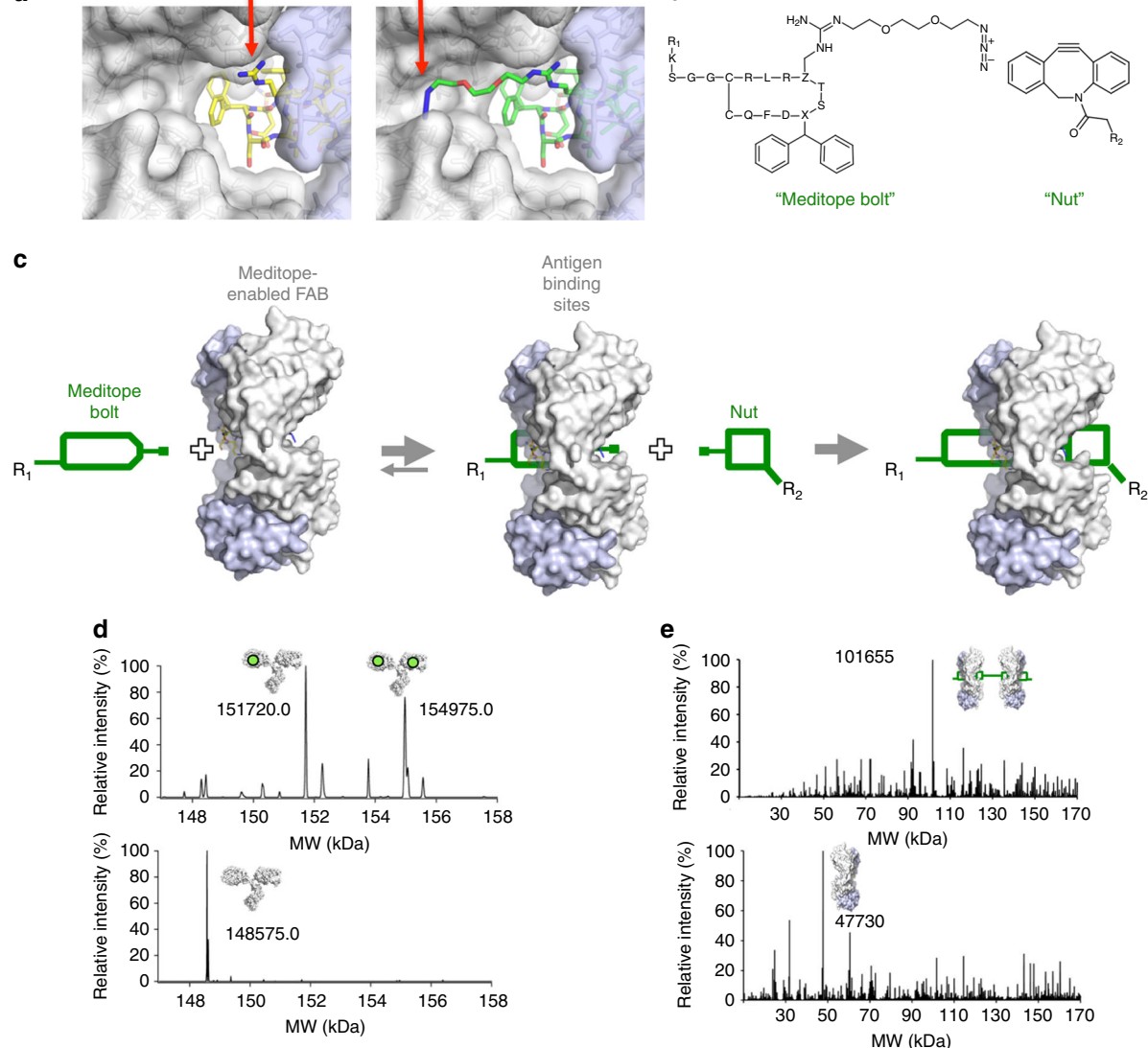

**Fig. 2** Development of a mechanical bond to interlock the meditope to a Fab. **a** Back view of Ile83Glu anti-HER2 antibody Fab with (Ac)CQFDA(Ph)$_2$STRRLKC meditope (left) and 5-diphenylalanine-8-Arg-(PEG)$_2$-azido-meditope (right). Light blue surface is the light chain; white surface is the heavy chain. The red arrows indicate the modified residue (arginine 8, left panel) and the reactive azide (right panel). **b** Chemical structures of the meditope bolt (meditope-containing reactive group) and nut (reactive steric group), the sequence is represented by single letter amino acid code where X represents 5-diphenylalanine and Z represents the modified azido-arginine. R1 and R2 represent potential modification sites. **c** Proposed scheme for forming a mechanical bond. After binding of the azido-meditope, the reactive azide on the backside of the Fab can react with a reactive cycloalkyne to form a mechanical bond. **d** Native mass spectrometry analysis of the mechanically interlocked meditope–AF647/Ile83Glu anti-HER2 memAb. The difference in molecular weight (3145 and 6400 Da) corresponds to 1:1 or 2:1 complex of mechanically interlocked meditope:IgG (top panel). The spectrum for unlabeled IgG is shown in the bottom panel. The experimentally determined mass of meditope–DIBO–AF647 is 3293.3 Da (see Supplementary Fig. 4). **e** Mass spectrometry analysis of "molecular dumbbell". Top panel: mass spectrum of purified reaction product; calculated mass: 101,634 (2×47730 + 2×2178 + 1818), observed mass: 101,655. Bottom panel: mass spectrum of the Ile83Glu, meditope-enabled Fab used for the reaction above; calculated mass: 47720, observed mass: 47730

Reagent K (5 mL) was used for cleavage (3 h 40 min). The material was purified by semi-preparative HPLC, and pure fractions were combined. A disulfide bond was formed by oxidative cyclization using 20 mL 2.5% NH4OAc pH 6–7 with 20% dimethylsulfoxide (DMSO). The reaction was allowed to proceed for ~24 h and then purified using semi-preparative HPLC, affording 2.6 mg azido-meditope variant a (MS: $[M + H]^{3+}$ calculated: 712.0072, observed: 712.0090). Azido-meditope variant b was synthesized similarly using compound b (variant b: MS: $[M + H]^{3+}$ calculated: 726.6826, observed, 726.6844). Structures of meditope variants a and b are in Supplementary Methods.

**Peptide conjugation.** Ac-KSADASK peptide was conjugated with DBCO-(PEG)$_4$-NHS in 100 mM NaHCO$_3$ buffer 1 h at room temperature. After purification by RP-HPLC, the product was lyophilized and analyzed by mass spectrometry (MS: $[M + 2 H]^{2+}$ - calculated MW 908.93; observed MW 908.94) and HPLC.

Cys-His$_6$ peptide was conjugated with sulfo DBCO-(PEG)$_4$-maleimide in PBS solution at room temperature for 1 h. After purification by RP-HPLC, the product was analyzed by HPLC and mass spectrometry (MS: $[M + 2 H]^{2+}$– calculated MW 885.34; observed MW 885.26), and lyophilized.

**Antibody expression and purification.** Both original and meditope-enabled anti-HER2[1], M5A[6], OKT3, and pertuzumab (Supplementary Methods) light and heavy chain genes were purchased from Atum in pJ609 and pJ607 expression vectors, respectively. They were sub-cloned into vector 253074 (Atum) for expression. Ile83Glu substitution on the light chain of the meditope-enabled anti-HER2 mAb to increase meditope-binding affinity was generated using standard site-directed mutagenesis and verified by DNA sequencing.

mAbs were produced using the ExpiCHO Expression System (Gibco). The vectors were co-transfected into ExpiCHO cells at 1 μg total plasmid DNA per 6

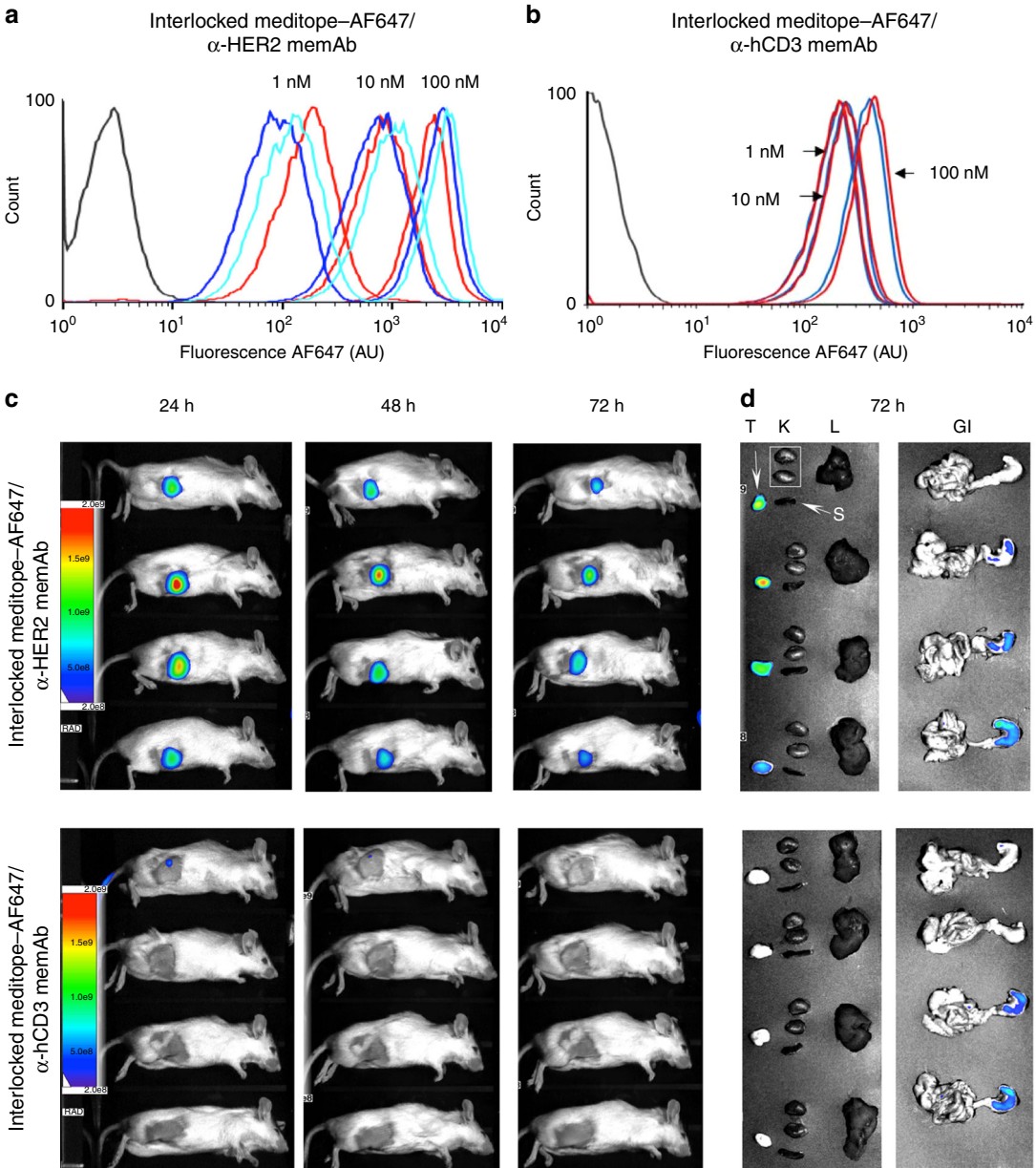

**Fig. 3** In vivo imaging. **a** Analytical flow cytometry analysis of BT474 HER2 overexpressing breast cancer cells incubated with 1, 10, or 100 nM of Ile83Glu anti-HER2 memAb with 5-diphenylalanine meditope–AF647 at a 1:1 molar ratio (blue), Ilu83Glu anti-HER2 memAb directly labeled with AF647 (green) or with mechanically interlocked 5-diphenylalanine-8-Arg-(PEG)$_2$-azido-meditope–DIBO–AF647 Ile83Glu anti-HER2 memAb (red). **b** Analytical flow cytometry analysis of PBMCs incubated with 1, 10, and 100 nM labeled antibodies. The red trace is the mechanically interlocked, 5-diphenylalanine-8-Arg-(PEG)$_2$-azido-meditope–DIBO–AF647 Ile83Glu anti-CD3 memAb (based on Okt3). The blue traces are cells treated with the anti-CD3 memAb with pre-bound 5-diphenylalanine-meditope-Alexa-647 (e.g., non-covalent). The black trace represents untreated cells. **c** Xenogen images of mice bearing BT474 xenograft tumors and injected through tail vain with mechanically interlocked meditope–AF647–Ile83Glu anti-HER2 memAb top row, n = 4) or mechanically interlocked meditope–AF647–Ile83Glu anti-CD3 memAb (control, bottom row, n = 4). Images were taken 24, 48, and 72 h after injection. **d** Tissue distribution of mechanically interlocked AF647–meditope–Ile83Glu anti-HER2 memAb and the mechanically interlocked AF647–meditope–Ile83Glu anti-CD3 memAb 3 days post-injection: tumors (T) and organs (K kidneys, L liver, S spleen, GI gastrointestinal tract)

million cells. At 18 h post-transfection, ExpiFectamine, CHO Enhancer, and ExpiCHO Feed were added to the flask and then transferred to a 32 °C incubator (Max Titer Protocol). The culture supernatant was collected at 14 days post-transfection. The mAbs were purified from the culture supernatants using Protein G agarose beads (Genscript) and further purified on a Superdex HiLoad 16/60 75 or 200 pg column (GE Healthcare). The sized mAbs were concentrated and buffer-exchanged into PBS with Amicon Ultra-15 centrifugal filter units (Millipore) and stored at 4 °C.

**Fab purification**. Ile83Glu anti-HER2 Fab was generated with immobilized papain (Pierce) according to manufacturer's protocol, followed by Q-sepharose chromatography and SEC purification. Purity was verified by SDS-PAGE[1].

**Protein crystallization and structure determination**. Crystals of apo Ile83Glu anti-HER2 memAb and its complexes with (Ac)CQFDA(Ph)$_2$STRRLRCGGSK, 5-diphenylalanine-8-Arg-(PEG)$_2$-azido-meditope or 5-diphenylalanine-8-Arg-(PEG)$_3$-azido-meditope were obtained by hanging drop method by mixing an equal volume of protein and precipitant (100 mM Tris, pH 7.2, 16–18% PEG 3350 and 150–200 mM NaCl). Apo Ile85Glu anti-HER2 memAb was mixed with equimolar concentration of Protein A and Protein L to a final concentration of 80 µM. Peptides were added to a final ratio of 1:10 of Fab:peptide. Crystals were cryoprotected in the precipitant solution with the addition of 20% meso-erythritol and frozen in a stream of cold nitrogen (100 K). Diffraction data were collected on a Rigaku Micromax X-007 HF with RAXIS IV++ detector at 100 K, processed with XDS[24] and refined with Phenix[25] with iterative

model building in Coot[26]. Final structures have 98.4% (5UED), 99.1% (5U5F), 98.6% (5U6A), and 99.0% (5U5M) of Ramachandran favored, and no Ramachandran outliers.

**Surface plasmon resonance**. Kinetic binding assays of peptides and memAb Fabs were obtained by SPR analysis on a Biacore T100 instrument. Ligands (memAb variants and sHER2) were prepared in 10 mM sodium acetate pH 5.5 (GE Healthcare Life Sciences) for covalent coupling onto series S CM5 sensor chips (GE Healthcare Life Sciences) using amine coupling chemistry at densities estimated to result in $R_{max}$ values between 50 and 150 RU using the equation:

$$R_L = R_{max} \times (\text{ligand MW}/\text{analyte MW}) \times 1/S_m.$$

Analyte samples were prepared as 2-fold serial dilutions in HBS-EP+ buffer (GE Healthcare Life Sciences) and were flowed over the immobilized ligand surface at 30 μL min$^{-1}$. The running buffer in all experiments was HBS-EP+ and the regeneration buffer was 10 mM glycine pH 2.0 followed by a wash with HBS-EP+. Binding constants and kinetic rate constants were calculated using the 1:1 binding model in BiaEvaluation software.

**Click chemistry**. In a typical reaction, IgG or Fab (0.15 mM) was incubated in 1× PBS with azido-meditope (0.9 mM) at a 2.2:1 or 1.1:1 ratio, respectively for 15 min at room temperature followed by the addition of four molar excess of reactant (DBCO-peptide, 10 mM; DIBO-Alexa647, 1.6 mM). The reaction was allowed to proceed for 2–16 h at 25 °C and the final products were separated on PD10 columns (GE Healthcare Life Sciences) and further purified on a Superdex 75 or 200 Increase 10/300 GL column (GE Healthcare Life Sciences). Ile83Glu memAb anti-HER2 Fab with interlocked meditope–DBCO-His$_6$ was generated in the same manner and purified from unreacted Fab using Ni-NTA beads (Qiagen). Unreacted DBCO-His$_6$ was removed on a Zeba Spin desalting column with 7000 MW cutoff (ThermoFisher) and analyzed by native PAGE (Supplementary Figure 2).

Interlocked Ile83Glu anti-HER2 Fab with 5-diphenylalanine-8-Arg-(PEG)$_2$-azido-meditope–(Cy5)$_2$–DBCO–magnetic beads was obtained as follows: 30 μL of DBCO–magnetic beads slurry (~0.33 mM DBCO) were separated from the solution using a magnetic separation rack (NEB) and washed 4 times with 50 μl of PBS, and resuspended in 30 μL of PBS. Fab was incubated with the azido-meditope at a 1.2:1 molar ratio for 15 min. Fab/meditope complex was mixed with DBCO-magnetic beads to a final molar ratio of 1.2:1:0.9. The admixture was allowed to mix overnight at room temperature in the dark. Beads were washed 10× with 50 μL of PBS before being resuspended in 60 μL of PBS. The presence of Fab in solution and bound to the beads was assayed by Western blot using an α-human κ light chain-HRP antibody (AbCam). Control with Fab (to account for non-specific binding of Fab to the beads) was performed in an analogous manner.

To generate a Fab-DBCO-(PEG)$_4$-AcKSADASK-(PEG)$_4$-DBCO-Fab "molecular dumbbell," pre-formed Fab-5-diphenylalanine-8-Arg-(PEG)$_3$-azido-meditope complex (0.4 mM, 15 min at room temperature at 1:0.9 molar ratio of Fab:azido-meditope) was allowed to react overnight at room temperature with 0.5 molar equivalent of 10 mM DBCO-(PEG)$_4$-AcKSADASK -(PEG)$_4$-DBCO and was purified on a Superdex 75 10/300 Increase column (GE Healthcare Life Sciences) and analyzed by mass spectrometry. Mass spectrometry data was acquired using a Waters Synapt G2G mass spectrometer and MassLynx software

**Stability of the interlocked meditope IgG complex**. For SEC experiments, the interlocked meditope Ile83Glu anti-HER2 Fab complex (starting volume 150 μL) was dialyzed against PBS (500 mL) in a dialyzer with an MWCO 7 kDa membrane. Buffer was changed at 62, 86, 110, 134, 159, and 230 h. Samples were assayed by SEC (S75 10/300 Increase) at 0, 62, 110, 159, 232, and 278 h.

For flow cytometry studies, the interlocked meditope–Ile83Glu–anti-HER2 IgG was diluted in PBS to 5 μM in 20 μL. The solution was dialyzed as described above, at 37 °C on an orbital shaker for 11 days with the PBS changed on days 2, 5, and 8. Samples were withdrawn after 1, 3, and 10 days and analyzed by flow cytometry with BT474 cells[6].

**Animal experiments**. All mouse experiments were approved by City of Hope Institutional Animal Care and Use Committee (IACUC 91037). Animal handling was conducted in accordance with ethical standards and regulations by IACUC. Twenty mice were used for the study to ensure enough mice bearing tumors of similar size. Each cohort included four mice (a typical sample size was $n \geq 3$ per cohort), no randomization or blinding was used for this study. NOD/SCID/Il-2rg (NSG) female mice (approximately 9 weeks old, Jackson Laboratory) were intra-muscularly injected with Delestrogen (0.8 mg/0.25 mL, estradiol valerate) 2 days before being subcutaneously injected in the shoulder or low flank with $4 \times 10^6$ mycoplasma-negative BT474 cells suspended in 1% human serum albumin in Hanks Balanced Salt Solution (HBSS) and then mixed to a 1:1 ratio with matrigel (BD) a total volume of 200 μL. Tumor xenografts were allowed to establish for 28 days, and confirmed by palpation (100 mm$^3$ minimum tumor size). Interlocked AF647–meditope–anti-HER2 IgG or interlocked AF647–meditope–OKT3 IgG (100 μg in 200 μL saline) was administered through tail vein injection in four mice.

Mice were imaged at 24, 48, and 72 h post-injection using a Lago system (Spectral Instruments Imaging) with 640 nm excitation and 690 nm emission filters. For image acquisition, mice were sedated with isoflurane for approximately 5 min. Mice were euthanized after the 72-h time point and tumors and major organs (liver, kidneys, spleen, and tumor) were harvested. The tumors and organs were then imaged on the Lago system using the same filter sets.

**Data availability**. All protein structures (5U3D, 5U5F, 5U6A and 5U5M) were deposited in the Protein Data Bank (http://www.rcsb.org). All relevant data are available from the authors.

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

## Acknowledgements

We gratefully acknowledge support from the Alicia and John Kruger Gift (J.C.W. and D. A.H.), the Leo and Anne Albert Charitable Trust (J.C.W. and G.S.), W.M. Keck Medical Foundation (J.C.W. and D.A.H.), the Carl and Roberta Deutsch Foundation (J.C.W. and G.S.), and awards R21 CA135216 and R21 CA174608 from the National Cancer Institute (J.C.W.). Research reported in this publication included work performed in the Animal Resource Center, X-ray Crystallography, and Mass Spectroscopy and Proteomics Cores supported by the National Cancer Institute of the National Institutes of Health under award number P30CA033572 (PI, Steven Rosen). The content is solely the responsibility of the authors and does not necessarily represent the official views of the National Institutes of Health. Finally, we thank Kurt Jenkins, Victor Kenyon, and Kaniel Cassady as well as the past and present members of the Williams and Horne groups for technical support and helpful comments, Desiree Crow from the Small Animal Imaging Core, and Roger Moore from the Mass Spectrometry Core for their technical expertise.

## Author contributions

J.C.W., K.P.B., J.W.P., G.S., and D.A.H. conceived the study. K.P.B., J.W.P., C.Z., J.X., Y. M., J.D.K., L.H.G, and K.N.A. generated reagents, performed the experiments, and analyzed the data. K.P.B. coordinated the data collection. D.C. oversaw tumor-imaging studies. J.C.W. and K.P.B. co-wrote the paper. All authors contributed to the review of the manuscript.

## Additional information

**Competing interests:** J.C.W. and D.A.H. have multiple patents (issued and pending) covering the meditope technology, are co-founders, shareholders, and members of the SAB of Meditope Biosciences, Inc. However, all of the work reported here was funded by charitable contributions to City of Hope and federal grants (i.e., none of the research reported here was supported by Meditope Biosciences, Inc.). The remaining authors declare no competing interests.

