## [Peer Review File · Nature Communications]

Reviewers' comments:

Reviewer #1 (Remarks to the Author):

The manuscript of Bzymek and colleagues describes the generation of antibody/Fab-meditope complexes with substantially improved stability. This is achieved using several stepwise approaches, including the insertion of a functional group on the mediotope that allows it to be locked onto the Fab by addition of a steric group. The mediotope-Fab complex retains antigen binding, and the utility of mediotopes for the imaging of tumors in mouse xenograft models is demonstrated. This study represents an important advance for the application of mediotopes, since it overcomes earlier limitations due to instability of mediotope-Fab/antibody association. I have the following comments:

1. The numbers of experiments that were carried out for the tumor imaging need to be stated: if they were only done once with two mice, then this needs to be repeated to ensure reproducibility. Also, there is clear signal outside the tumor, which is inconsistent with the statement 'signal was found exclusively in tumors and not in any other organs.'
2. Related to the imaging, it would also be useful to show the distribution of a control mediotope-antibody (that does not bind to tumor antigen) in tumor-bearing mice.
3. Supplementary Figure 6. The data should also be presented as mean fluorescence intensity vs. time for each complex.
4. The labeling in supplementary Figure 7c is not clear. Which plots does the '100 nM' refer to?
5. A more general point is that the numbers of experiments/repeats need to be stated for the flow cytometry and other data.
6. The methods for tumor implantation, mouse strain etc. are missing.

Minor comments:

1. 'FACS' should be replaced by 'flow cytometry'.
2. There are multiple grammatical errors that need to be corrected.

Reviewer #2 (Remarks to the Author):

Concerning the click chemistry section, the authors should provide more experimental details, in particular:

- 1) the concentrations of IgG or Fab used in the click reaction and the solvent are not specified.
- 2) the range of molar excess of DBCO-peptide/DIBO-Alexa647 (2-8 molar) and the reaction time interval (2-16 hours) are too broad. The importance of these parameters should be discussed.

In general, the approach described is feasible and the results convincing. However, I don't feel qualified to evaluate this manuscript in all its parts.

Reviewer #3 (Remarks to the Author):

Nice work.

Comments and Suggestions

1. Not sure why a chemical bond is called a "mechanical bond" in the abstract. Interlocking is certainly an apt description.
2. The mediotope sequence should be cited in the text. Right now, it can only be found in the

legend for Figure 1.

3. Not sure how to interpret Figure 1b; what do DLE, LE, L, and Ori mean?
4. What residues are in green in "QFD" and what was changed to '5-diphenyl' in Figure 1c? Are green residues for the meditope or both the meditope and Fab?
5. Has there been an alanine-scan or series of truncations to identify the minimal ligand sequence and length, respectively, in the meditope?
6. What is the difference between the two panels in Figure 2a? Why are the meditopes in yellow or green. It would be nice if colors were consistent in the manuscript.
7. Figure 2c is very useful and informative. Clear cut SPR results.
8. I see that the meditope has 'supershifted' the Fab species in Supplementary Figure 2, but do not understand why there are two bands in each lane.
9. The authors may wish to cite alternative strategies for labelling the N-terminus of Cterminus of proteins [Schmidt M, Toplak A, Quaedflieg PJ & Nuijens T (2017) Enzyme-mediated ligation technologies for peptides and proteins. *Curr Opin Chem Biol* 38, 1–7] as well as a review of bioconjugation strategies [Hu QY, Berti F & Adamo R (2016) Towards the next generation of biomedicines by site-selective conjugation. *Chem Soc Rev* 45, 1691–1719.]
10. Obviously functionalization occurs in vitro. What are the pros and cons of the chemical reaction (i.e., cost, scalability, efficiency, potential for unwanted denaturation, aggregation, etc.)?

Reviewer #4 (Remarks to the Author):

The authors report on the enhancement of affinity and lifetime of a recently discovered mediotope-Fab interaction. I cannot comment on the overall novelty and impact of the manuscript because it is outside my expertise, but as requested I will comment specifically on the SPR measurements.

SPR measurements to establish the association and dissociation rate were performed on a commercial SPR instrument. The SPR measurements seem to have been done properly, but I miss an estimate of the error in the measurement of the rates and affinity. SPR measurements are sensitive to details such as antibody orientation on the substrate, temperature drift, contamination etc. Especially for low rate constants this can cause variability in the measured affinity. Although the supplementary information includes an error in Fig S8, this seems to be the fitting error and not an error that indicates variations between different measurements under the same conditions. To establish the significance of the observed changes in affinity this error should be experimentally determined at least for selected conditions/modifications.

Minor comments:

- Regarding Fig 1b: a number of datapoints are plotted that show the association and dissociation rate for different modifications. It is unclear what all the datapoints represent because only a few are labelled. The authors should indicate which datapoints represent which modifications, or perhaps better, only include the most relevant modifications in the graph to improve readability.
- Regarding Fig 1c:
 - o Different curves are plotted for each modification, it is not indicated which concentrations these curves represent, and whether the extracted rates are concentration dependent.
 - o The association rate is extremely fast compared to the time-scales plotted. Because large changes in association rate are at the origin of some of the observed changes in affinity it would be instructive to plot the short times separately so the changes in the on-rate are visible.

Reviewers' comments:

Reviewer #1 (Remarks to the Author):

The manuscript of Bzymek and colleagues describes the generation of antibody/Fab-meditope complexes with substantially improved stability. This is achieved using several stepwise approaches, including the insertion of a functional group on the mediotope that allows it to be locked onto the Fab by addition of a steric group. The mediotope-Fab complex retains antigen binding, and the utility of mediotopes for the imaging of tumors in mouse xenograft models is demonstrated. This study represents an important advance for the application of mediotopes, since it overcomes earlier limitations due to instability of mediotope-Fab/antibody association. I have the following comments:

1. The numbers of experiments that were carried out for the tumor imaging need to be stated: if they were only done once with two mice, then this needs to be repeated to ensure reproducibility.

We completely agree and have repeated the experiment with the recommended control. To this end, we have successfully reproduced the *in vivo* imaging studies data using 8 additional mice, all bearing tumors using BT474 cells (which overexpress Her2.) Four mice were treated with mechanically-interlocked, AF647-meditope-trastuzumab complex. The other four were treated with a mechanically-interlocked, isotype control memAb (we used a mediotope-enabled, anti-CD3 mAb based on Okt3 which does not cross react with murine CD3). As before, the interlocked AF647-meditope-trastuzumab accumulated at the tumor whereas "very weak; to no signal" was observed for the control. We have added these data to the manuscript and the methods. We also moved the original imaging studies to the supplementary data. We are grateful for the reviewer's suggestion including the control.

Also, there is clear signal outside the tumor, which is inconsistent with the statement 'signal was found exclusively in tumors and not in any other organs.'

We apologize for the confusing language. The original statement was intended to refer to the signal found in harvested organs 8 days post administration of the mechanically interlocked mediotope-IgG complex. In repeating the imaging experiment, we harvested the organs at three days post injection. This time we also harvested the GI tract. As shown in figure 3d, the signal is predominantly in the tumor. There was no signal in the other organs consistent with the first imaging studies (liver, kidneys and spleen). However, there is some signal in the stomach for the experimental and control group. This is not uncommon – dyes and imaging groups with different molecular composition and charge distributions are processed differently and accumulate in different tissues. We were told by several investigators that the signal in the stomach could be from chlorophyll in the animal chow as well. In future studies, we will fully characterize and alter the properties of interlocked mediotope to optimize the signal-to-noise (particularly to the tissue type – liver cancer surgeons would benefit from kidney excretion as opposed to GI excretion - personal communication with our collaborator Dr. Bouvet at UCSD as well as the co-author Dr. Singh, both pancreatic surgeons.)

2. Related to the imaging, it would also be useful to show the distribution of a control mediotope-antibody (that does not bind to tumor antigen) in tumor-bearing mice.

As noted above, this was done. Thank you for the suggestion – this is a far better control than we originally used.

3. Supplementary Figure 6. The data should also be presented as mean fluorescence intensity vs. time for each complex.

We have included a plot with mean fluorescence intensity vs. time for each complex, as recommended by the reviewer.

4. The labeling in supplementary Figure 7c is not clear. Which plots does the '100 nM' refer to?

We have corrected this. We apologize for the confusion caused by the errant labeling.

5. A more general point is that the numbers of experiments/repeats need to be stated for the flow cytometry and other data.

The number of experimental determination for each method has been included in the figure legends or the text.

6. The methods for tumor implantation, mouse strain etc. are missing.

This information has been added to the methods section under its own subsection.

Minor comments:

1. 'FACS' should be replaced by 'flow cytometry'.

We apologize for the loose language - we have changed it following reviewer's recommendation

2. There are multiple grammatical errors that need to be corrected.

We have used an in-house, professional editor to edit the manuscript for grammar and spelling errors.

Reviewer #2 (Remarks to the Author):

Concerning the click chemistry section, the authors should provide more experimental details, in particular:

- 1) the concentrations of IgG or Fab used in the click reaction and the solvent are not specified.
- 2) the range of molar excess of DBCO-peptide/DIBO-Alexa647 (2-8 molar) and the reaction time interval (2-16 hours) are too broad. The importance of these parameters should be discussed.

We agree that the reaction details were lacking in the original manuscript. We also admit that we have simply followed the literature and different manufacturer's recommendations for click reactions (Click Chemistry Tool and Invitrogen). We have not delved deeply into the optimization of the click reaction for several reasons. First, our primary goal was to establish that we could interlock the mediotope onto the Fab. To achieve this, we had to synthesize the azido-arginine derivative de novo and incorporate this azido-arginine into the mediotope peptide. Thus, we did not produce these azido-mediotopes at large scale and did not have enough material to 'fully' explore/optimize the reaction conditions. Second, we repeated the reaction multiple times for the biophysics, analytical cytometry, and imaging studies under slightly different conditions (2 hrs versus overnight). We didn't observe differences in the product that warranted a more thorough approach. This observation partly reflects the 'robustness' of the click chemistry, but also suggests that further optimization for efficiency (if necessary) is likely going to require significant chemical modifications on both sides including the presentation of the azide (e.g., further extension from the hole, more rigid linker between the guanidinium group of Arg8 and the azide, etc.) and strained alkyne (e.g., more strain or less strain of the alkyne, attachment position of the functional group, sterics etc.). We argue that exploring these parameters is beyond the focus of this manuscript, but certainly something we intended to explore in on-going studies. These studies will also explore different orthogonal pairs including tetrazine/strained alkenes and thiol groups.

However, directly to the reviewer's request, we added to the manuscript and the supplementary methods the conditions of a "typical reaction," providing the concentrations of the reagents, solvent, temperatures and time.

In general, the approach described is feasible and the results convincing. However, I don't feel qualified to evaluate this manuscript in all its parts.

Reviewer #3 (Remarks to the Author):

Nice work.

We very much appreciate the reviewer's positive comment!

Comments and Suggestions

1. Not sure why a chemical bond is called a "mechanical bond" in the abstract. Interlocking is certainly an apt description.

We understand – there is some confusion about the nomenclature. We originally called the clicked mediotope complex a 'locked-on' mediotope. In a brief conversation with Peter Dervan, he indicated the locked-on mediotope is referred to as a mechanical bond. Additional searches and perhaps more definitively, Stoddard and his contemporaries use "mechanically interlocked molecule". We have revised the manuscript to use interlocked where appropriate.

2. The meditope sequence should be cited in the text. Right now, it can only be found in the legend for Figure 1.

We agree and have added the meditope sequence in the introduction.

3. Not sure how to interpret Figure 1b; what do DLE, LE, L, and Ori mean?

We apologize for not defining these in the original manuscript – Ori is the original meditope, L is the Leucine → Diphenylalanine mutation, etc. Each label is now defined in the legend.

4. What residues are in green in "QFD" and what was changed to '5-diphenyl' in Figure 1c? Are green residues for the meditope or both the meditope and Fab?

Again, we apologize for the 'lab' language finding its way into the manuscript. (e.g., QFD being the first three amino acids after the disulfide bond in the original meditope and 5-diphenyl indicating the diphenylalanine modification at the fifth position.) We have addressed this either by defining acronyms or writing out the entire name for each throughout the text.

5. Has there been an alanine-scan or series of truncations to identify the minimal ligand sequence and length, respectively, in the meditope?

Yes, we and others have conducted extensive structure-affinity studies on cetuximab that have been published in the past few years (Donaldson *et al.*, Proc Natl Acad Sci U S A. (2013) 110(43), 17456-61; Bzymek *et al.*, Acta Cryst. (2016) F72, 434-442; Bzymek *et al.*, Acta Cryst. (2016) F72, 820-830; van Rosmalen *et al.*, J Biol Chem. 2017 Jan 27;292(4):1477-1489.)

6. What is the difference between the two panels in Figure 2a? Why are the meditope in yellow or green. It would be nice if colors were consistent in the manuscript.

We have revised the color scheme somewhat. We are using the change in color to indicate a modification in the meditope from the previous step. We hope this addresses your concern.

7. Figure 2c is very useful and informative. Clear cut SPR results.

Thank you!

8. I see that the meditope has 'supershifted' the Fab species in Supplementary Figure 2, but do not understand why there are two bands in each lane.

In our early studies of antibodies in general, we were very concerned about stability (asn modification, met oxidation, sugar modifications, etc.) as we observed additional bands from commercial sources. However, literature searches, communication with antibody experts at major meetings and our own experience indicates that it is common for antibodies undergo different modification over short and long periods that leads to heterogeneity (Dakshinamurthy *et al.*, Biologicals (2017) 46, 46-56; Khawli *et al.*, MAbs (2010) 2, 613-624 and references therein). Despite the 'general' acceptance of these modifications, we have run a native gel on Fab from herceptin obtained from the clinic (lane 1 below); an older preparation of Fab featured in this study (4 years old, lane 2), and a fresh preparation of Fab from this study (lane 3). In each case multiple bands are present, suggesting some modification of the antibody. Of note, we used freshly prepared Fabs and IgGs for all of our experiments.

9. The authors may wish to cite alternative strategies for labelling the N-terminus of C-terminus of proteins [Schmidt M, Toplak A, Quaedflieg PJ & Nuijens T (2017) Enzyme-mediated ligation technologies for peptides and proteins. *Curr Opin Chem Biol* 38, 1–7] as well as a review of bioconjugation strategies [Hu QY, Berti F & Adamo R (2016) Towards the next generation of biomedicines by site-selective conjugation. *Chem Soc Rev* 45, 1691–1719.]

Thank you! We have included these citations in the manuscript.

10. Obviously functionalization occurs in vitro. What are the pros and cons of the chemical reaction (i.e., cost, scalability, efficiency, potential for unwanted denaturation, aggregation, etc.)?

We very much appreciate the reviewer's question. We were/are hesitant to speculate too much in the manuscript about the future applications / benefits as we have been admonished by reviewers in previous manuscript submissions for doing so (e.g., we redacted a significant amount of the discussion in our PNAS manuscript describing the discovery and grafting of the meditope interaction). Having said that, we anticipate a number of scenarios where this approach could find clinical and 'practical' benefit over current conjugation technologies.

First, we need to further explore the parameters for efficient formation of the mechanically interlocked bond before we can truly calculate the cost, scalability, and physical parameters of this approach technology and contrast it to existing approaches. However, we have observed that the presence of the 5-diphenylalanine meditope improves the thermal stability of the Fab (often greater than 10 °C). Increased thermal stability of mAbs typically correlates to reduced aggregation and better shelf life (cold chain issues). Also, since the meditope is synthesized it is possible to systematically alter properties of the meditope outside the grafted binding site to alter PK/PD properties (increase the charge, the hydrodynamic radius, etc.). Our experience thus far suggests that it is more cost efficient to directly synthesize a short XTEN/PASylation sequence (to increase hydrodynamic radius) to the meditope than add an equivalent sized polyethylene glycol. In addition, it is straightforward to add the capthesin protease site, valine/citrulline, which is used for toxin release (e.g., Adcetris, Seattle Genetics), to the meditope during its synthesis. Overall, the technology may facilitate and provide advantages over current approaches. We expect these answers to become clear as we advance the technology.

Beyond these practical aspects, the interlocked meditope technology is inherently a combinatoric method approach. We can generate a dozen or more meditopes bearing different functional loads (toxins, dyes, nanoparticles, therapeutic peptides, etc.) and another dozen or more strained alkynes with different (or the same) functional groups. These can be mixed and matched with any meditope-enabled Fab or mAb to produce 100s of unique

compounds. With an appropriate read out, it is possible to rapidly identify lead compounds and/or study new 'biology.'

In terms of therapeutics, the interlocked meditope technology could solve a complication that can and has showed up in nonhuman, primate studies: Namely, immunogenicity of a conjugated biologic. Many methods used to conjugate drugs to antibodies (thiomAbs, unnatural amino acids, etc.) leave a "scar" within the linear peptide sequence of the mAb. These modified residues have the potential to be displayed by the MHCII of APCs and could be immunogenic. As an example, incorporation and administration of TNF α or EGF biologics bearing of p-nitrophenylalanine, sulfotyrosine or 3-nitrotyrosine amino acids induced a polyclonal IgG response in otherwise 'self' molecules (PMID: 21768354). The interlocked meditope is not chemically associated with the mAb. While this does not preclude an immune response to the interlocked meditope itself, not being directly associated with the antibody likely reduces the potential for antigen spread within an otherwise self 'molecule' (which can lead to serious conditions - see <https://academic.oup.com/ndt/article/18/7/1257/1809787>). Moreover, based on the extensive structural data we argue that it will be possible to identify and rapidly rectify potential immunogenicity associated with the meditope, if observed in non-human primate models. We are currently exploring which are the best methods to address this possibility.

As with all technologies, there are some cons. First, the azido-arginine derivatives are not commercially available. While we will make them available to other investigators when possible, we will have to convince a supplier to make the azido-arginine or obtain a small RO3 to produce sufficient amount for distribution. As noted above, we will further explore reaction conditions and other bio-orthogonal chemistry approaches to better understand and identify the best conditions to efficiently 'lock-on' the meditope. In addition, another con is that the meditope binding site must be grafted on to the antibody of interest. We do not feel like this is a significant hurdle. Most antibodies are now being produced recombinantly. Furthermore, the DNA is often codon optimized. It is possible to add the meditope binding site at this point. Thus far, we have not observed significant differences in expression levels between parental and meditope-antibodies. The current list of meditope-enabled mAbs stands over 50 in our lab. Finally, we note that nearly every other method to conjugation mAbs (thiomAbs, sortase, FGE, etc.) also must genetically engineer the mAb.

Reviewer #4

SPR measurements to establish the association and dissociation rate were performed on a commercial SPR instrument. The SPR measurements seem to have been done properly, but I miss an estimate of the error in the measurement of the rates and affinity. SPR measurements are sensitive to details such as antibody orientation on the substrate, temperature drift, contamination etc. Especially for low rate constants this can cause variability in the measured affinity.

We thank the reviewer for thorough analysis of our SPR results. And all the points made here are generally valid. However, the overriding objective of the meditope and antibody modifications was to increase the lifetime of the interaction. The half-life of the complex is given by $\tau_{1/2} = k_{off}/\ln(2)$. More importantly, k_{off} is a first order reaction and thus independent of the analyte concentration (i.e., the units are s^{-1}). We argue that most of these concerns raised here (orientation, temperature drift, and contamination) are highly sample dependent (e.g., protein-protein interactions or conjugation of a unique reactive lysine at an interface of interest.) We have not observed dramatic differences in the meditope/Fab system (the meditope binding site is buried and the meditope is relative small compared to an antigen interface. We also do not observe significant differences in the calculated values if we flow over a given meditope when the lane is conjugated with a Fab or the full IgG. In addition, our calculated values for the K_D and k_{on}/k_{off} for trastuzumab binding to the extracellular domain of Her2 are similar to the values reported in the literature. Finally, all the reagents used here were of highest purity available, all chips were prepared in the same manner, and all experiments had proper controls (e.g. original meditope for meditope variant studies.) We also agree with the reviewer that slow off-rates are close to the limit of the instrument. In fact, the long dissociation rates is precisely what drove us to do many of latter measurements at 37 °C (as opposed to 20 to 25 °C typically reported by others.) This point is stated in the text. Thus, while the reviewer correctly points out that orientation, temperature drift, etc. can affect these measurements, they likely produce a small correction in this system (<10%).

Although the supplementary information includes an error in Fig S8, this seems to be the fitting error and not an error that indicates variations between different measurements under the same conditions. To establish the significance of the observed changes in affinity this error should be experimentally determined at least for selected conditions/modifications.

All of the measurements presented in Supplementary Table 1 were fit using global analysis was performed using the software as described in the methods section. However, the reviewer is correct. Many of the interactions were measured as a single SPR run ($n=1$) as we did not observe a significant improvement in the off-rate for that particular variant compared to the previous variant. Based on extensive studies with the meditope-Fab system, we have yet to see a 'tenfold' difference in the off-rate based on multiple measurements using multiple chips using the same meditope-enabled mAb from different preparations (Donaldson, J. M. et al. PNAS 110, 17456-17461, doi:10.1073/pnas.1307309110 (2013); Bzymek, K. P. et al. Acta Cryst. Section F, 72, 820-830, doi:10.1107/s2053230x16016149 (2016); Bzymek, K. P. et al. Acta Cryst. Section F, 72, 434-442, doi:10.1107/s2053230x16007202 (2016) and more.). Due to time and financial constraints (producing second and third batches of Fab, second and third batches of each peptide, and generating fresh chips and instrument time), we moved on. However, for the final modifications that did enhance the off-rate, SPR measurements were performed ($n=4$) (with different Fabs/Mabs and peptide preparations and are presented as the average with standard deviation (see Supplementary Table 1).

Minor comments:

- Regarding Fig 1b: a number of data points are plotted that show the association and dissociation rate for different modifications. It is unclear what all the datapoints represent because only a few are labelled. The authors should indicate which datapoints represent which modifications, or perhaps better, only include the most relevant modifications in the graph to improve readability.

The figure is a simple visual comparison of the on-rates/off-rates for each variant used in this study. The identity of each point can be determined using Supplementary Table 1. We continue to feel that the original graph is useful as it shows some meditope variants have similar K_D 's but much different k_{on}/k_{off} 's. Moreover, the half-life, $\tau_{1/2}$, is the parameter we were trying to maximize.

- Regarding Fig 1c:

Different curves are plotted for each modification, it is not indicated which concentrations these curves represent, and whether the extracted rates are concentration dependent. The association rate is extremely fast compared to the time-scales plotted. Because large changes in association rate are at the origin of some of the observed changes in affinity it would be instructive to plot the short times separately so the changes in the on-rate are visible.

We understand the reviewer's comments, but note that it is difficult to 'visually judge' the on-rate from the plots alone as it depends on the concentration of the analyte. As the concentrations of the analyte are adjusted (after a preliminary SPR run) to optimize the signal and fitting (e.g., weak interactions need more analyte to get the signal), the on-rate curve will "look" faster or slower. The value from the fit, which is globally performed over all analyte concentrations used, is given in Supplementary Table 1. We show the SPR traces in 1c, which include the fits, to indicate that the calculated values are consistent with the data.

REVIEWERS' COMMENTS:

Reviewer #3 (Remarks to the Author):

Thanks for responding thoroughly to the reviewers' comments.

Reviewer #4(Remarks to the Author):

Although I cannot judge all aspects of the rebuttal, the authors have addressed most of my concerns.

Regarding comment 5.1 (estimate of the measurement error) it is important to include at least a rough error estimate in the manuscript or supporting information to give quantitative significance to the reported affinities. The estimate can be along the same lines as described in the rebuttal.

With this change I recommend publication of this nice piece of work.

Reviewer #3 (Remarks to the Author):

Thanks for responding thoroughly to the reviewers' comments.

We are very grateful to have the comments as they substantially improved the manuscript.

Reviewer #4 (Remarks to the Author):

Although I cannot judge all aspects of the rebuttal, the authors have addressed most of my concerns.

Regarding comment 5.1 (estimate of the measurement error) it is important to include at least a rough error estimate in the manuscript or supporting information to give quantitative significance to the reported affinities. The estimate can be along the same lines as described in the rebuttal.

With this change I recommend publication of this nice piece of work.

We agree and have added the estimated error in the supplementary data.